# The two-qubit singlet/triplet measurement is universal for quantum computing given only maximally-mixed initial states

Terry Rudolph [1] ✉ & Shashank Soyuz Virmani [2] ✉

In order to delineate which minimalistic physical primitives can enable the full power of universal quantum computing, it has been fruitful to consider various measurement based architectures which reduce or eliminate the use of coherent unitary evolution, and also involve operations that are physically natural. In this context previous works had shown that the triplet-singlet measurement of two qubit angular momentum (or equivalently two qubit exchange symmetry) yields the power of quantum computation given access to a few additional different single qubit states or gates. However, Freedman, Hastings and Shokrian-Zini[1] recently proposed a remarkable conjecture, called the 'STP=BQP' conjecture, which states that the two-qubit singlet/triplet measurement is quantum computationally universal given only an initial ensemble of maximally mixed single qubits. In this work we prove this conjecture. This provides a method for quantum computing that is fully rotationally symmetric (i.e. reference frame independent), using primitives that are physically very-accessible, naturally resilient to certain forms of error, and provably the simplest possible.

Since the origin of quantum computation, it has been of fundamental interest to understand which types of physical operations enable universality. Beyond the standard textbook gate model of quantum computation, it was soon realised that measurements could be used not only for readout of information but also as dynamical elements. The most widely known example is perhaps the scheme of Raussendorf and Briegel[2], wherein given a particular fixed many-particle entangled state (the 'cluster state') a quantum computation can be executed by adaptively performing (destructive) single-qubit measurements. Originally it was envisaged that the cluster state would be pre-generated by multi-qubit entangling unitary operations. In light of such, one may wonder whether one can push the role of measurements yet further still, completely eliminating any use of coherent unitary operations.

The first scheme that required no unitary operations at all was proposed by Nielsen[3]; it required being able to perform multiple distinct (non-destructive) 4-qubit measurements. This was simplified in various ways in subsequent works. Fenner and Zhang[4] provided a scheme using multiple different 2- and 3-qubit measurements. Leung[5] proposed a scheme with multiple different 2-qubit measurements or a single 4-qubit measurement. Perdrix[6] devised a method using three different single-qubit measurements and only additional measurement of the two-qubit observable $X \otimes Z$ ($X, Z$ refer to the standard Pauli operators). At about the same time, the universality of 2-qubit (destructive) 'fusion' measurements, physically natural for photonic qubits, was proven[7]. Note that in some of these works additional assumptions were needed on the initial single-qubit sources available.

Partly motivated by a desire to develop quantum computational schemes using more physically natural measurements, in ref. 8 we constructed a quantum computational scheme based upon the measurement of two-qubit total angular momentum. More explicitly, this measurement—which we refer to as the singlet/triplet measurement or s/t measurement (following the notation of ref. 1)—consists of only two

[1]Department of Physics, Imperial College London, London SW7 2AZ, UK. [2]Department of Mathematics, Brunel University London, Kingston Ln, London, Uxbridge UB8 3PH, UK. ✉e-mail: tez@imperial.ac.uk; shashank.virmani@brunel.ac.uk

outcomes, one corresponding to the projector

$$P_s := |\Psi^-\rangle\langle\Psi^-| \tag{1}$$

onto the singlet state $|\Psi^-\rangle := (|01\rangle - |10\rangle)/\sqrt{2}$, and the other corresponding to the 'triplet' projector onto its orthogonal complement:

$$P_t := I - P_s, \tag{2}$$

(which projects onto the subspace spanned by the 'triplet' of states $|00\rangle, |11\rangle, (|01\rangle + |10\rangle)/\sqrt{2}$). Note that because

$$(U \otimes U) P_s (U^\dagger \otimes U^\dagger) = P_s \tag{3}$$

for any single-qubit unitary $U$, this measurement has the important property, relevant to later discussion, that it is rotationally invariant, i.e., invariant under local changes of basis/reference frame. The manifest rotational and permutational invariance of the singlet/triplet projectors underpins their generic physical naturalness - these subspaces will often be energetically separated, even in physical systems with degrees of freedom unrelated to the total angular momentum of spin-1/2 particles.

While the scheme of ref. 8 used only the s/t measurement for all dynamical and readout purposes, an additional necessary assumption was that input qubits could be prepared in at least three single-qubit (possibly mixed) states $\rho_a, \rho_b, \rho_c$ with linearly independent Bloch vectors. Rotational invariance then implies that s/t measurements are universal given only an initial qubit mixed state of the form

$$\int dU \, U^{\otimes 3N} (\rho_a^{\otimes N} \otimes \rho_b^{\otimes N} \otimes \rho_c^{\otimes N}) U^{\dagger \otimes 3N}, \tag{4}$$

where the integral represents a uniform average, over the Haar measure $dU$, of all possible single-qubit unitaries $U$ performed identically on every qubit in the system. Here $3N$, the total number of qubits involved in implementing the computation, grows only polynomially in the underlying algorithm size.

The universality of s/t measurements under other assumptions was reconsidered recently by Freedman, Hastings and Shokrian-Zini[1], in which the authors proposed a remarkable conjecture, referred to as the 'STP=BQP conjecture', namely that s/t measurements alone are universal given essentially arbitrary input states. In ref. 1, the authors posed the conjecture by formally assuming that the inputs are a supply of singlet states. However, it is clear that the inputs could be chosen in many other ways. For instance, maximally mixed qubits are an equivalent choice as they can be produced from singlet states by discarding one particle, and can produce singlet states when measured with the s/t measurement. Moreover, as s/t measurements can create singlet states from almost every input source, the only exception being inputs of such high symmetry they return only (or mostly) triplet outcomes, the STP=BQP conjecture in fact implies that the s/t measurement is quantum universal given almost any input states.

In this paper, we show that the STP=BQP conjecture is in fact true. We do so by demonstrating that replacing Eq. (4) with even a resource of only maximally mixed single-qubits suffices to make the s/t measurement universal. We choose to assume that the inputs are maximally mixed, as these are the only rotationally invariant single-qubit states, and this choice renders our scheme more obviously minimalistic.

## Results

In this section, we give an overview of the main ingredients of the proof, including some required elements from previous works[1, 8]. Technical details are deferred to the 'Methods'.

## Prior work

First let us briefly summarise the parts of refs. 1, 8 that we will need. In refs. 1, 8, various ways were proposed for using s/t measurements to perform quantum computation using additional single-qubit resources, which may be either states[8] or unitaries[1]. In any such scheme, it is clear that as the s/t measurement is the only multiparticle operation, it must be the resource that is used to build multiparty entanglement. Further, as the s/t measurement gives probabilistic outcomes, this has to be done offline, so that we only use the entanglement once we are satisfied that it has been created to a sufficient quality.

In ref. 8, this was achieved by building cluster states[2], using the triplet outcome of the s/t measurement to fuse smaller entangled clusters into bigger ones, having initially started from entanglement created by the singlet outcome. To execute the computation, single particle measurements were constructed by using s/t measurements and ancilla qubits prepared in known states along the desired measurement axis. In both building the cluster states and executing the measurements, it was initially assumed in[8] that there are supplies of highly pure single-qubit states. However, as shown in detail in ref. 8, this assumption could be relaxed significantly to assume only the input ensemble of Eq. (4). Hence the ensemble (4), and s/t measurements on arbitrary pairs of qubits, are sufficient for universal quantum computation.

In ref. 1, the computational power of s/t measurements was considered in other contexts, with the aim of proposing and providing supporting evidence for the STP=BQP conjecture. In addition to demonstrating quantum universality when the s/t measurement is supplemented by single-qubit $X, Z$ gates[1], demonstrated that the s/t measurement alone is at least as powerful as the weak model of permutational quantum computation[9], and with the addition of post selection, it is equivalent to post-BQP. A sampling problem was also proposed that by definition could be efficiently solved using s/t measurements, in spite of suggestions that it might not be possible classically efficiently.

A key primitive proposed in ref. 1, which we will also make use of in the present work, is the implementation of (an exponentially good) measurement of the total angular momentum by using only repeated pairwise s/t measurements.

## Proof strategy

We show that the STP=BQP conjecture is indeed correct by following the approach of our previous work[8], but dropping the assumption that we have been given the resource (4). We show instead how such a state may be approximately created efficiently using only s/t measurements acting upon maximally mixed input qubits.

The construction proceeds in two steps, a *maximally mixed symmetric state preparation* step ('MMSS preparation'), and a *relative localisation step* (Fig. 1).

To understand these steps, let us ask ourselves how we might go about creating the state (4)? It is clear that we cannot produce (say) $N$ copies of a particular pure state $|\psi\rangle^{\otimes N}$ just using our available resources, as $|\psi\rangle^{\otimes N}$ is not invariant under an arbitrary $U^{\otimes N}$, whereas the s/t measurements and maximally mixed states are. However, we might consider trying to produce the following state, which we call a *maximally mixed symmetric state* ('MMSS'):

$$\rho_N^{sym} := \int dU \, U^{\otimes N} \left( |\psi\rangle\langle\psi|^{\otimes N} \right) U^{\dagger \otimes N}. \tag{5}$$

This state has an ensemble interpretation of comprising $N$ copies of an unknown pure state, and it is invariant under arbitrary $U^{\otimes N}$. From standard results in the theory of symmetric group representations and/or the theory of quantum angular momentum, it is also equal to the maximally mixed state on the subspace of symmetric states, hence the symbol $\rho_N^{sym}$. The first step of our argument—the *MMSS preparation*

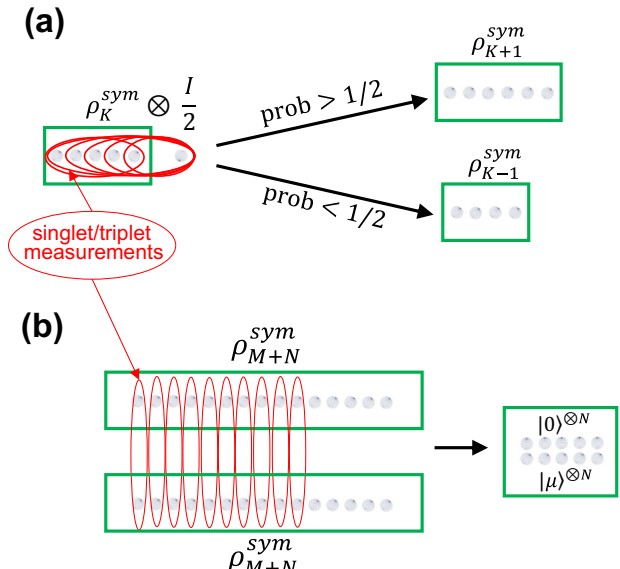

**Fig. 1 | The two steps of the protocol. a** *MMSS preparation*: Bringing up a maximally mixed qubit to an MMSS $\rho_K^{sym}$ of $K$ qubits, and measuring its total angular momentum using a procedure from[1] implements a biased random walk that probabilistically generates an MMSS of increasing size. **b** *Relative Localisation*: Begin with two MMSS states of $M + N$ qubits. Each can be interpreted as a random mixture of many copies of an unknown pure state. By pairing up and measuring $M$ qubits from each using the s/t measurement (red ovals) we build up information about the relative angle the Bloch vectors of these unknown pure states make against each other. This allows us to estimate the unknown pure states to high accuracy, up to an unimportant global unitary transformation. The remaining qubits can hence be used, together with s/t measurements, to implement cluster state quantum computation, as detailed in ref. 8.

step—is to show how we can efficiently prepare a state that is a good approximation to $\rho_N^{sym}$. We do this by starting with two maximally mixed qubits and measuring them with an s/t measurement. With probability 3/4 the qubits are left in the state $\rho_2^{sym}$. We then add maximally mixed qubits one at a time, and each time we do this we perform a total angular momentum measurement using a procedure described in ref. 1. We will see that if this measurement yields the right outcome then we successfully transform $\rho_n^{sym} \rightarrow \rho_{n+1}^{sym}$ to high accuracy, whereas if it fails we transform $\rho_n^{sym} \rightarrow \rho_{n-1}^{sym}$ to high accuracy. It turns out that probabilities of each outcome are such that this 'random walk' reaches our target $\rho_N^{sym}$ with only a polynomial cost. We note that this protocol is very similar in spirit to the protocols of section 2 of ref. 1, and it is possible that an appropriate variant of their *splitting protocol* could be used to create $\rho_N^{sym}$ more efficiently (the splitting protocol is a primitive in which given an initial multi-qubit state of total angular momentum $S$, one divides the qubits into two sets $A, B$ and uses *s/t* measurements to efficiently realise with high accuracy any desired local total angular momenta $S_A, S_B$ of each subset consistent with the constraint $|S_A - S_B| \leq S \leq S_A + S_B$, and parts of the protocol target the generation of symmetric subspace states on certain subsystems).

In the second step, "relative localisation" we see how to transform three copies of $\rho_{N'}^{sym}$ into a good quality copy of (4) for an $N'$ that need only be polynomially larger than $N$. This is done using *measurement-induced relative localisation*, in full analogy with a similar phenomenon for optical phase, Bose-Einstein condensate phase, and particle position studied in detail in ref. 10. To understand what we mean by *relative localisation*, consider two input copies of $\rho_{N'}^{sym}$. Each can be interpreted as a supply of (generically different) unknown pure states, say $|a\rangle$ and $|b\rangle$. If we were to take a qubit in $|a\rangle$ and a qubit in $|b\rangle$ and measure them with a s/t measurement, then the probability of getting a triplet

outcome is given by:

$$\frac{1 + |\langle a|b\rangle|^2}{2} \tag{6}$$

The probability of getting a triplet therefore tells us the angle between the Bloch vectors of the two unknown states $|a\rangle$ and $|b\rangle$. Hence, we may take two copies of $\rho_{N'}^{sym}$ and repeatedly take one qubit from each source, measuring them using s/t measurements. The observed frequency of triplet outcomes will allow us to estimate the angle between the Bloch vectors to high precision. Once we are satisfied with the statistical precision that we have reached, we stop and use the remaining unmeasured qubits for computation. They can be considered to be in a state that is a good approximation of

$$\int dU \, U^{\otimes 2N} \left( |a\rangle\langle a|^{\otimes N} \otimes |b\rangle\langle b|^{\otimes N} \right) U^{\dagger \otimes 2N} \tag{7}$$

for some arbitrary $|a\rangle$, $|b\rangle$, with $0 < |\langle a|b\rangle|^2 < 1$ determined by the observed frequency of triplet outcomes. Creating (4) follows exactly the same process, but starting with three copies of $\rho_{N'}^{sym}$ rather than two. Up to picking an arbitrary handedness of our coordinate system, it turns out that the frequency of triplet outcomes observed between the three sources allows us to create (4) with high accuracy at polynomial cost. This allows us to conclude that the STP=BQP conjecture is indeed correct.

We note that our constructions are unlikely to be optimal, and it is likely that further work could make them significantly more efficient.

## Discussion

If we consider operations needed to build a quantum computer from scratch (i.e., without a prior source of entanglement), it is clear that for quantum resources we at least need (i) a supply of qubits, (ii) at least one two-qubit operation to generate entanglement, and (iii) at least one binary outcome measurement so that the computation readout is humanly accessible. The STP=BQP theorem is noteworthy because it meets (ii) and (iii) with only a single two-outcome, two-qubit measurement, with no other dynamical operation or measurement needed. It is hence even more remarkable that it meets (i) in a manner almost completely agnostic about the initial state of the qubit resource: provided that we are promised that a sufficient number of singlet outcomes will occur, the singlet outcomes can be used to prepare the single-qubit maximally mixed states required.

The model is also minimal with respect to its use of rotationally invariant primitives. This could be of practical import for systems subject to collective decoherence.

From a foundational perspective, it also is interesting that the computation would be described identically, and using real numbers only, in every choice of reference frame.

Elaborating on this last point: consider two non-communicating parties observing a physical system performing a classical computation, and each writing down a mathematical description of such. It can reasonably be arranged that these descriptions are identical, perhaps up to an ambiguity about which physical state of a bit in the machine corresponds to mathematical 0, and which to 1.

The same is not true if the parties instead observe a device implementing a quantum computation via the standard circuit architecture. While some physical systems do have an intrinsic natural (say) $Z$ eigenbasis (e.g., right/left circular polarisation of a photon, ground/excited atomic states), agreeing on the $X$ eigenbasis (corresponding to agreeing on an orientation in space for polarisation, or origin of time for atomic energy levels) for all known physical qubits requires extra exchange of physical information to align the reference frame in question[11]. This lack of agreement could be annoying. As one example, if one observer does a universal computation using a simple (say, real-

valued) gate set the other would typically disagree and say that it was messy and complex-valued. Another is that because the parties disagree on the correct Pauli bases, procedures that would be manifestly fault-tolerant using the stabiliser formalism to one observer would not be so to the other.

By contrast, the scheme presented in this work has the following property: no matter how many such disagreeably misaligned observers there are, we can arrange for them to all describe a single universal quantum computation being performed in such a way that every single experimenter will at all times during the computation assign the exact same mathematical description in terms of states and operators to all elements of the computation. (Again up to an ambiguity as to which of the outcomes $P_{s/t}$ is assigned 0/1.) It is perhaps also philosophically interesting that this description would at all times remain real-valued.

While other schemes for achieving similar basis independence can be constructed using methods reviewed in ref. 12 on encoding and processing information in decoherence-free subsystems, they are considerably more complex than the methods of this work. For example, they require encoding in large multi-qubit states, more complicated unitary gates, and often make use of intermediate operations that are not, in fact, rotationally invariant.

We do not anticipate that our scheme is optimal in terms of resource scaling, nor is it explicitly fault-tolerant in its present form. In this regard it is worth noting that a considerably simpler approach than generation of increasingly large cluster states and subsequent simulation of single-qubit measurements (as was done in ref. 8) would probably be to implement fusion-based quantum computing[13]. In that approach, one need only show the ability to create small, constant-sized states and the ability to implement a Bell measurement (or one of its fusion variations). That approach also will automatically yield fault tolerance. A deeper analysis of such may merit further attention.

## Methods

In this section, we present details of the proof that STP=BQP. As described in the Results there are two steps to the procedure: the MMSS preparation step, and the relative localisation step. In the first subsection, we describe the MMSS preparation, and in the following two subsections we describe the relative localisation.

### MMSS preparation

The construction proceeds recursively. Suppose that we start with $K$ qubits prepared in $\rho_K^{sym}$. We bring in a new qubit in the maximally mixed state and randomly pick pairs of qubits to undergo a polynomial number of s/t measurements (more efficient choices than random pairings will certainly exist—for example, interpreting the switches of the networks in ref. 14 as s/t measurement locations). If we only ever find triplet outcomes, we perform an exponentially good approximate projection[1] into the state $\rho_{K+1}^{sym}$. This occurs with probability $P(K) = (K+2)/(2K+2) > 1/2$. If we ever find a singlet outcome we discard those two qubits and the remaining $K-1$ qubits are left in $\rho_{K-1}^{sym}$.

We can interpret the protocol as a 1-d random walk process where we begin at $K=1$, and have a probability $P(K)$ of stepping to the right, and $1-P(K)$ of stepping to the left. The boundary at $K=0$ is absorbing (fail, restart) and let us consider our target to be creating $\rho_N^{sym}$ for some fixed $N$. The solution to this problem can be found in ref. 15. Note that the particle must eventually be absorbed at one or the other of the boundaries. Eq. (2.7) of ref. 15 for our case yields that the probability it is absorbed at the right hand boundary is $(N+1)/2N$, i.e., slightly higher than 1/2. Thus we have finite probability of eventual success. To ensure the resources consumed (qubits/time steps) are polynomial, we need to compute the conditional mean for the number of steps before stopping (absorption at a boundary). This can be found by solving the recurrence relations (3.1)–(3.3) in ref. 15. For starting at $K=1$, we find

the expected number of steps before absorption is $(N^2 + 3N - 4)/6$, which grows polynomially with $N$.

### Relative localisation

The second step is to see how two sufficiently large maximally mixed symmetric states can be converted into an ensemble equivalent to a Haar−twirled product state over pure states with fixed overlap, i.e., a state of the form:

$$\int dU\, U^{\otimes 2N}\left(|a\rangle\langle a|^{\otimes N} \otimes |b\rangle\langle b|^{\otimes N}\right)U^{\dagger\otimes 2N} \tag{8}$$

for some arbitrary (but known) $|a\rangle$, $|b\rangle$ with $0 < |\langle a|b\rangle|^2 < 1$.

This can be done by using measurement-induced localisation of the relative angle between initial (mixtures of) spin coherent states, similar to the cases studied in ref. 10. Once we localise two such ensembles, we can use the same procedure to relationally localise further ensembles to the first two—we leave that analysis to the next subsection, and here consider only two ensembles.

The basic intuition is simple: we start with two sources $\rho_{N+M}^{sym} \otimes \rho_{N+M}^{sym}$, as created in the first step, and interpret each as an ensemble of $N+M$ copies of a randomly selected pure state. We pair up $M$ of the spins from each source and perform the singlet/triplet measurement on each pair, enabling us to get a good estimate of the overlap between the two (random) pure states. We then use the remaining $2N$ qubits for computation, under the assumption that the overlap is the estimated one (we remark that the relative localisation could possibly be induced more efficiently using approximate total angular measurement protocols of ref. 1 together with some form of global angular momentum inference scheme, see ref. 16). As we now demonstrate, a fixed overall error across the $2N$ qubits requires $M$ to grow only polynomially in $N$.

Because of the collective unitary freedom, we are free to decide that the first MMSS is actually a source of $|0\rangle^{\otimes N+M}$, and the second state $|\theta\rangle^{\otimes N+M}$ is specified by the relative angle $\theta \in [0, \pi)$ its Bloch vector makes with the first source state, where $\theta$ has p.d.f $\sin(\theta)/2$. An s/t measurement on $|0\rangle \otimes |\theta\rangle$ gives a triplet outcome with probability $q = (1 + \cos^2(\theta/2))/2 = (3 + \cos(\theta))/4$. The total probability over the $M$ measurements of obtaining $n_1$ triplet outcomes is

$$P(n_1) = \int_0^\pi d\theta \binom{M}{n_1} q^{n_1}(1-q)^{M-n_1}\frac{\sin(\theta)}{2} \tag{9}$$

$$= 2\binom{M}{n_1}\int_{1/2}^1 dq\, q^{n_1}(1-q)^{M-n_1} \tag{10}$$

This has the convenient interpretation that the probability of seeing a given number of triplets is described by a Bernoulli trial with a uniformly chosen $q$ in the interval $[1/2, 1]$. Estimating $\theta$ corresponds to estimating $q$ given the observed $M$, $n_1$, so we will also write $|q\rangle := |\theta\rangle$.

Considering the function

$$T(a, b) := \int_{1/2}^1 dq\, q^a(1-q)^b \tag{11}$$

$$= \frac{a!b!}{(a+b+1)!}\frac{1}{2^{a+b+1}}\sum_{j=0}^a \binom{a+b+1}{j} \tag{12}$$

we can use standard identities (en.wikipedia.org/wiki/binomial_coefficient) for partial sums of binomial coefficients to see that $T(a, b)$ is exponentially close (in $M = (a + b)$) to $\left(\binom{a+b}{a}(a+b+1)\right)^{-1}$

when $a > (a+b)/2$. Applying this to $P(n_1)$, we find that it is exponentially close to $2/(M+1)$, which means that with high probability on any given run of the procedure we will observe $n_1 > M/2$ triplet outcomes, and from now on, we consider only situations where this has occurred.

The probability density of $q$ given $n_1$ triplet outcomes is (over the domain $q \in [1/2, 1]$):

$$\Pr(q|n_1, M) = \frac{q^{n_1}(1-q)^{M-n_1}}{T(n_1, M-n_1)}, \tag{13}$$

from which we wish to bound the goodness of our estimated value of $q$ (and hence $\theta$). The mean and variance for this inference problem are given by

$$\mu = \frac{T(n_1+1, M-n_1)}{T(n_1, M-n_1)} \approx \frac{n_1+1}{M+2}$$
$$\sigma^2 = \frac{T(n_1+2, M-n_1)}{T(n_1, M-n_1)} - \mu^2 \approx \frac{(n_1+1)(M+1-n_1)}{(M+2)^2(M+3)}, \tag{14}$$

where $\approx$ denotes exponential closeness. A simple upper bound on the variance is then $\sigma^2 < 1/M$.

Now, we are roughly in the following situation: we will operate as if $q = \mu$, i.e., the state of the second $N$ qubits is $|\mu\rangle^{\otimes N}$ (by collective rotational freedom taken to be a state in the right semicircle of the XZ plane in the Bloch sphere), but with a low probability ($\le 1/h^2$ by the Chebyshev inequality (en.wikipedia.org/wiki/chebyshev's_inequality)) the actual value of $q$ could be further than $h\sigma$ from this. In later calculations, we will pick $h = M^{1/6}$. The error we want to understand will ultimately arise from the trace distance between the estimated state and the actual one, and so we wish to bound this.

To make things simpler we first ask, for any pair of $q_1, q_2$ with a fixed value of $|q_1 - q_2|$, what is the largest possible trace distance between the corresponding quantum states $|q_1\rangle$, $|q_2\rangle$? Elementary considerations yield:

$$\| |q_1\rangle\langle q_1| - |q_2\rangle\langle q_2| \| \le 2\sqrt{8|q_1 - q_2|} \tag{15}$$

This can be shown as follows. The state $|q\rangle$ has $z$ component of its Bloch vector given by $z = 4q - 3$, so a given value of $|q_1 - q_2|$ constrains the Bloch vectors of $|q_1\rangle$, $|q_2\rangle$ to have a projection on the $z$ axis to a fixed interval $4|q_1 - q_2|$. Positioning one end of this interval at either the north or south poles of the Bloch sphere yields the largest possible trace distance consistent with this projected value. We can use Eq. (15) to bound the overall error via:

$$\| |\mu\rangle\langle\mu|^{\otimes N} - \int dq \, \Pr(q|n_1, M)|q\rangle\langle q|^{\otimes N} \| \tag{16}$$

$$\le \int dq \, \Pr(q|n_1, M) \| |\mu\rangle\langle\mu|^{\otimes N} - |q\rangle\langle q|^{\otimes N} \| \tag{17}$$

$$\le \sqrt{N-1} \int dq \, \Pr(q|n_1, M) \| |\mu\rangle\langle\mu| - |q\rangle\langle q| \| \tag{18}$$

$$\le 2\sqrt{8(N-1)} \int dq \, \Pr(q|n_1, M)\sqrt{|q-\mu|} \tag{19}$$

$$\le 2\sqrt{8(N-1)}\sqrt{\int dq \, \Pr(q|n_1, M)|q-\mu|} \tag{20}$$

$$\le 2\sqrt{8(N-1)}\sqrt{\frac{1}{h^2} + h\sigma} \le 2\sqrt{8(N-1)}\sqrt{\frac{2}{M^{1/3}}} \tag{21}$$

where the first inequality is the triangle inequality, the second is because for pure states it holds that $\| \psi^{\otimes N} - \phi^{\otimes N} \| \le \sqrt{N-1} \| \psi - \phi \|$, the third is from the bound (15), the fourth is concavity of the square root, the fifth is from the largest probabilities (and $|q-\mu|$ values) consistent with the Chebyshev inequality, and the last is obtained by using $\sigma \le 1/\sqrt{M}$ and picking $h = M^{1/6}$.

We deduce that given the target overall error of $\epsilon$, we can choose $M \sim (N/\epsilon^2)^3$, which is a polynomial cost.

## Relatively localising a further source

We now describe how the errors arising from the relative localisation of a further source to the first two may be controlled. The method is essentially analogous to the discussion in the previous subsection, albeit with some modifications to control more complicated integrals. Let us begin by assuming that we have already taken two MMSS sources, and have relatively localised them. One source is (by the protocol) exactly $|0\rangle$, the other is $|\mu\rangle$, which is subject to statistical error. However, we will proceed as if it is exact, noting by our previous argument that the error introduced by assuming this can be made arbitrarily small at polynomial cost. Recall that $|\mu\rangle$ can be assumed to have Bloch vector components $x > 0$ and $y = 0$ (i.e., is in the positive $x$ direction of the XZ plane). Now we bring in a third MMSS, which we consider to be a source of a random state $|\psi\rangle$. This source will give a Bloch vector linearly independent from the other two sources almost surely. We will localise it relative to the other two sources by using triplet measurements. We will call this 'two source relative localisation', and refer to the previous relative localisation as 'single source relative localisation'. As $|0\rangle$ and $|\mu\rangle$ are in the XZ plane, the two source relative localisation will give us information on the $x$ and $z$ components of the Bloch vector of $|\psi\rangle$. As $|\psi\rangle$ is pure, the $y$ component will then be fixed up to a sign as $y = \pm\sqrt{1 - x^2 - z^2}$. We are free to pick one sign as that corresponds to choosing the handedness of our coordinate system, so we will assume that $y = +\sqrt{1 - x^2 - z^2}$. Denote the Bloch vectors of $|0\rangle$ and $|\mu\rangle$ by $(0, 0, 1)$ and $(\sin(\theta), 0, \cos(\theta))$ (where $\theta \in (0, \pi)$), respectively, and denote the Bloch vector of the random state $|\psi\rangle$ by $(x, y, z)$. The probabilities of getting triplet outcomes when measuring $|0\rangle \otimes |\psi\rangle$ and $|\mu\rangle \otimes |\psi\rangle$ are the random variables given by:

$$q_a = \frac{1 + |\langle 0|\psi\rangle|^2}{2} = \frac{3+z}{4}$$
$$q_b = \frac{1 + |\langle\mu|\psi\rangle|^2}{2} = \frac{3 + x\sin(\theta) + z\cos(\theta)}{4} \tag{22}$$

respectively. A pair $(q_a, q_b)$ is hence in one-to-one correspondence with $(x, z)$, and so through observed estimates of $(q_a, q_b)$, we will be able to estimate $|\psi\rangle$ (as we take $y = +\sqrt{1 - x^2 - z^2}$). If there were no connection between $|0\rangle$ and $|\mu\rangle$ there would be no correlation between $q_a, q_b$. However, because of Eq. (22), there will be restrictions on the possible values of $q_a, q_b$. Let $Q$ be the set $\{q_a, q_b | q_a \in [1/2, 1], q_b \in [1/2, 1]\}$ of all possible $q_a, q_b$ pairs, when we neglect correlations, and denote by $R \subset Q$ the subset of values of $q_a, q_b$ permitted by Eq. (22). Note $R$ depends on the value of $\theta$. However, we suppress this dependence as it will not play a significant role. Over $Q$ let us denote the p.d.f. of $q_a, q_b$ by $f(q_a, q_b)$ - although this will be zero on $Q \backslash R$, it is convenient to define it on the whole of $Q$. Again, $f(q_a, q_b)$ depends on $\theta$ but the precise details will not be needed. To simplify our computations we will later neglect the correlations between $q_a, q_b$, and perform inference as if they come from a product distribution on the whole of $Q$. This will allow us to utilise bounds computed for the single-source relative localisation. Even though this adds an additional layer of approximation, it allows relatively straightforward bounds on

error to be computed, and the overall error incurred can still be made arbitrarily small at polynomial cost.

We begin by constructing for the two source case, a bound similar to Eq. (15). First consider two pure states $|\psi_1\rangle$, $|\psi_2\rangle$ with Bloch vectors $(x_1, y_1 = +\sqrt{1-x_1^2-z_1^2}, z_1)$ and $(x_2, y_2 = +\sqrt{1-x_2^2-z_2^2}, z_2)$ respectively. Consider the projections of the Bloch vectors in the $XZ$ plane (i.e., $(x_1, z_1)$ and $(x_2, z_2)$), these projections have a separation $l = \sqrt{(x_1-x_2)^2 + (z_1-z_2)^2}$. What is the largest possible trace distance between the two pure states, consistent with a given value of $l$? It is not difficult to show that the solution is the same as given in Eq. (15), i.e.,

$$\| |\psi_2\rangle\langle\psi_2| - |\psi_1\rangle\langle\psi_1| \| \leq 2\sqrt{8}l \qquad (23)$$

It is convenient to derive an upper bound to $l$ utilising $q_a, q_b$ as our coordinates instead of $x, z$. Given two pure states $|\psi_1\rangle$, $|\psi_2\rangle$ represented by $(q_a, q_b)$ and $(q_a + \Delta q_a, q_b + \Delta q_b)$, respectively, let us bound the value of $l$. We note that we can write Eq. (22) as:

$$\begin{pmatrix} q_a \\ q_b \end{pmatrix} = \frac{1}{4}\begin{pmatrix} 0 & 1 \\ \sin(\theta) & \cos(\theta) \end{pmatrix}\begin{pmatrix} x \\ z \end{pmatrix} - \frac{3}{4}\begin{pmatrix} 1 & 0 \\ 0 & 1 \end{pmatrix} \qquad (24)$$

Define a matrix $A$ such that

$$A^{-1} = \frac{1}{4}\begin{pmatrix} 0 & 1 \\ \sin(\theta) & \cos(\theta) \end{pmatrix} \qquad (25)$$

($A$ is well-defined under our assumption that $\theta \in (0, \pi)$). Denoting the operator norm of $A$ by constant $c$ (while this depends upon $\theta$, in this stage of the relative localisation we are treating $\theta$ as a constant), and using the triangle inequality, we find that

$$l \leq c|\Delta q_a| + c|\Delta q_b|. \qquad (26)$$

We then may put this together with Eq. (23), and use the triangle inequality once more, to give:

$$\| |\psi_2\rangle\langle\psi_2| - |\psi_1\rangle\langle\psi_1| \| \leq 2\sqrt{8c}\left(|\Delta q_a|^{1/2} + |\Delta q_b|^{1/2}\right) \qquad (27)$$

It will be convenient to use this inequality is it separates contributions from errors in estimates of $q_a$ and errors in estimates of $q_b$, and this will allow us to straightforwardly apply the single-source analysis. Let $n_a, n_b$ be the number of triplet outcomes observed when localising to $M$ copies of $|0\rangle$, and $M$ copies of $|\mu\rangle$, respectively. Let (omitting the '$dq_a dq_b$' in this and all subsequent double integrals to keep the notation uncluttered)

$$\Pr(q_a, q_b|n_a, n_b, M) := \frac{f(q_a, q_b)q_a^{n_a}(1-q_a)^{M-n_a}q_b^{n_b}(1-q_b)^{M-n_b}}{\int_Q f(q_a, q_b)q_a^{n_a}(1-q_a)^{M-n_a}q_b^{n_b}(1-q_b)^{M-n_b}} \qquad (28)$$

be the probability density of the state being described by $q_a, q_b$ conditioned upon observing $n_a, n_b$ triplets when measuring against $M$ copies of $|0\rangle$ and $M$ copies of $|\mu\rangle$. As we will be neglecting correlations between $q_a$ and $q_b$ when estimating them in our two source relative localisation, we use the single-source estimates. Consequently, let $\mu_a, \mu_b$ be the mean values of $q_a, q_b$, and $\sigma_a^2, \sigma_b^2$ the corresponding variances, as constructed in Eq. (14) for the single-source case. Let $\sigma = \max\{\sigma_a, \sigma_b\} \leq 1/\sqrt{M}$. Let $|\mu_a, \mu_b\rangle$ be the pure state corresponding to setting $q_a = \mu_a$ and $q_b = \mu_b$ exactly, and $|q_a, q_b\rangle$ be the pure state corresponding to setting $q_a$ and $q_b$ exactly. In analogy to the single-source relative localisation,

we have (omitting steps that are essentially identical to the previous case):

$$\| |\mu_a, \mu_b\rangle\langle\mu_a, \mu_b|^{\otimes N} - \int_Q \Pr(q_a, q_b|n_a, n_b, M)|q_a, q_b\rangle\langle q_a, q_b|^{\otimes N} \| \qquad (29)$$

$$\leq \int_Q \Pr(q_a, q_b|n_a, n_b, M) \| |\mu_a, \mu_b\rangle\langle\mu_a, \mu_b|^{\otimes N} - |q_a, q_b\rangle\langle q_a, q_b|^{\otimes N} \| \qquad (30)$$

$$\leq 2\sqrt{8(N-1)c}\int_Q \Pr(q_a, q_b|n_a, n_b, M)|q_a - \mu_a|^{1/2} + \text{sim } b \qquad (31)$$

$$\leq 2\sqrt{8(N-1)c}\sqrt{\int_Q \Pr(q_a, q_b|n_a, n_b, M)|q_a - \mu_a|} + \text{sim } b \qquad (32)$$

We would hence like to control expressions like:

$$\int_Q \Pr(q_a, q_b|n_a, n_b, M)|q_a - \mu_a| \qquad (33)$$

where $\Pr(q_a, q_b|n_a, n_b, M)$ is given by Eq. (28). Intuitively, it is clear that while the conditional probability $\Pr(q_a, q_b|n_a, n_b, M)$ is correlated across $q_a, q_b$, after doing many observations we should still expect the posterior distribution on $q_a, q_b$ to become strongly peaked around $\mu_a, \mu_b$ anyway, and so our estimates adapted from the single-source scheme should still be close. Let us show that this is indeed the case with high enough probability. Let $C_h \subset Q$ be the 'close' set $\{q_a, q_b|q_a \in [\mu_a - h\sigma, \mu_a + h\sigma], q_b \in [\mu_b - h\sigma, \mu_b + h\sigma]\}$ of $q_a, q_b$ values that are within $h\sigma$ of the estimates, and $F_h = Q\backslash T_h$ be the 'far' set of $q_a, q_b$ values that are more than $h\sigma$ from the estimates, where $\sigma$ is the maximum of the two variances. We will pick the value of $h$ later. Let us also define the product measure $\text{Prod}(q_a, q_b|n_a, n_b, M) := \Pr(q_a|n_a, M)\Pr(q_b|n_b, M)$ that would arise if there were no correlation (i.e., if we were relatively localising to two sources that are independent of each other, using the single-source scheme). Explicitly:

$$\text{Prod}(q_a, q_b|n_a, n_b, M) = \frac{(\frac{1}{4})q_a^{n_a}(1-q_a)^{M-n_a}q_b^{n_b}(1-q_b)^{M-n_b}}{\int_Q (\frac{1}{4})q_a^{n_a}(1-q_a)^{M-n_a}q_b^{n_b}(1-q_b)^{M-n_b}} \qquad (34)$$

(although the inclusion of the 1/4 in both the numerator and denominator is superfluous, we include it as it is the analogue of $f(q_a, q_b)$). Let us define the following integrals:

$$C = \int_{C_h} f(q_a, q_b)q_a^{n_a}(1-q_a)^{M-n_a}q_b^{n_b}(1-q_b)^{M-n_b} \qquad (35)$$

$$F = \int_{F_h} f(q_a, q_b)q_a^{n_a}(1-q_a)^{M-n_a}q_b^{n_b}(1-q_b)^{M-n_b} \qquad (36)$$

From the definitions of these integrals, we have that

$$\int_Q \Pr(q_a, q_b|n_a, n_b, M)|q_a - \mu_a| \leq \frac{h\sigma C + F}{C + F}$$
$$\leq h\sigma + \frac{F}{C} \leq \frac{h}{\sqrt{M}} + \frac{F}{C} \qquad (37)$$

To control this error, we need to first need to upper bound $F/C$. We will do this by first computing an upper bound to $F$ and a lower

bound to $C$. For $F$ we first note that:

$$F \leq \max_{F_h}\left(q_a^{n_a}(1-q_a)^{M-n_a} q_b^{n_b}(1-q_b)^{M-n_b}\right) \tag{38}$$

$$= \max_{F_h}(2^{-MH(\vec{n}_a|\vec{q}_a)-MH(\vec{n}_a)-MH(\vec{n}_b|\vec{q}_b)-MH(\vec{n}_b)}) \tag{39}$$

where the relative entropy $H(\vec{n}_a|\vec{q}_a)$ and Shannon entropy $H(\vec{n}_a)$ are constructed from the probability distributions defined by

$$\vec{n}_a := \left(\frac{n_a}{M}, 1-\frac{n_a}{M}\right) \tag{40}$$

$$\vec{q}_a := (q_a, 1-q_a) \tag{41}$$

with similar definitions for $b$. We may now appeal to Pinsker's inequality (en.wikipedia.org/wiki/pinsker's_inequality) to lower bound the relative entropy. Together with the fact that $H(\vec{n}_a), H(\vec{n}_b) \geq 0$ we obtain:

$$F \leq 2^{-M\min_{F_h}\left(\|\vec{n}_a-\vec{q}_a\|^2+\|\vec{n}_b-\vec{q}_b\|^2\right)/2} \tag{42}$$

where the norm represents the 1-norm. To a high degree of approximation $\mu_a = n_a/M$ and $\mu_b = n_b/M$, and so from our definition of $F_h$, this becomes:

$$F \leq O\left(2^{-4Mh^2\sigma^2}\right). \tag{43}$$

Now let us turn to lower bounding $C$. Let us tentatively assume that $\min_{C_h} f(q_a, q_b) > k$, for some constant $k > 0$ (which up to some relatively mild restrictions we will be able to choose). We will later discuss why we may make this assumption with high enough probability. Hence we have:

$$C \geq 4k \int_{C_h} \left(\frac{1}{4}\right) q_a^{n_a}(1-q_a)^{M-n_a} q_b^{n_b}(1-q_b)^{M-n_b} \tag{44}$$

We now note that the integral in this lower bound is the probability, under the product distribution, of each of $q_a, q_b$ being within $h\sigma$ of their means. Exploiting the Chebyshev inequality (applied independently to both parts of the product distribution) we hence get

$$C \geq 4k\left(1-\frac{1}{h^2}\right)^2 \tag{45}$$

Putting the upper bound on $F$ together with the lower bound for $C$ gives

$$\int_Q \Pr(q_a, q_b|n_a, n_b, M)|q_a - \mu_a| \leq \frac{h}{\sqrt{M}} + \frac{O(2^{-4Mh^2\sigma^2})}{4k\left(1-\frac{1}{h^2}\right)^2} \tag{46}$$

If we now pick, for example, $h = M^{1/4}$, then the previous bound becomes

$$\int_Q \Pr(q_a, q_b|n_a, n_b, M)|q_a - \mu_a| \leq O(M^{-1/4}) + O(2^{-4M^{1/2}}) \tag{47}$$

As with the single-source case, this leads to a polynomial overhead for any desired target error.

What remains is demonstrating that we may pick a suitable constant $k > 0$. For a given (large enough) value of $M$ and observed values of $n_a, n_b$ we would like the resulting set $C_h$ (which is fixed by $n_a, n_b$ and

by our choice of $h = M^{1/4}$) to be such that $\min_{C_h} f(q_a, q_b) > k$. For a given $k$ consider the upper level set $L_k$ of $f(q_a, q_b)$, i.e., $L_k := \{(q_a, q_b)|f(q_a, q_b) \geq k\}$, so our requirement can be re-expressed as the requirement $C_h \subset L_k$. Assume that we have picked a $k$ such that $L_k$ is of non-zero size. For a small constant tolerance $\epsilon > 0$ that we shortly choose, consider the subset $W_k^\epsilon \subset L_k$ of $(q_a, q_b) \in L_k$ such that for $\Delta := h\sigma + \epsilon$, the neighbourhood $(q_a - \Delta, q_a + \Delta) \times (q_b - \Delta, q_b + \Delta)$ is contained in $L_k$. As the upper bound $h\sigma \leq 1/M^{1/4}$ is not increasing in $M$, we can assume (say by assuming we do not consider $M$ less than some large enough constant) that this target subset $W_k^\epsilon$ is of constant size. So with constant probability the values of $q_a, q_b$ that are realised will fall within $W_k^\epsilon$. The values of $n_a/M \approx \mu_a, n_b/M \approx \mu_b$ will converge to these $q_a, q_b$ exponentially quickly in $M$, and so by choosing the tolerance $\epsilon$ to accommodate this, with constant probability the observed $n_a, n_b$ will be such that $C_h \subset L_k$. If the observed $n_a, n_b$ do not satisfy $C_h \subset L_k$, we may simply abandon and repeat until we get a suitable $n_a, n_b$.

## Data availability
No data were generated or used in the work.

## Code availability
No code was generated or used in the work.

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

## Acknowledgements

S.S.V. acknowledges thought-provoking discussions with Nihaal Virmani. The authors thank Imperial College London Open Access Fund for funding the open access costs for this article.

## Author contributions

T.R. and S.S.V. contributed equally to this work.

## Competing interests

The authors declare no competing interests.
