## [Peer Review File · Nature Communications]

The two-qubit singlet/triplet measurement is universal for quantum computing given only maximally-mixed initial statesREVIEWER COMMENTS

Reviewer #1 (Remarks to the Author):

The purpose of this paper is to prove that a certain model of measurement-based quantum computation is universal for BQP, thereby resolving an open problem of Freedman, Hastings, and Shokrian-Zini. The model considered is one where the initial state of n qubits is maximally mixed, and where the only allowed operations are measurements that project onto singlet/triplet subspaces. The authors say that this is foundationally interesting, because it shows that universal QC is possible using states and operations that are "always real in every reference frame."

Is this "just another BQP-universality result" for some particular set of operations, alongside dozens of others that are now known? Or is it indeed of foundational interest and therefore suitable for Nature Communications? Alas, I didn't feel like I even understood the setup well enough to be able to answer that question with confidence. Here were my main issues:

(1) What ARE singlet/triplet measurements? Are they the measurements that project onto

$|00\rangle+|11\rangle$, $|00\rangle-|11\rangle$, $|01\rangle+|10\rangle$, $|01\rangle-|10\rangle$

as well as

$|000\rangle+|111\rangle$, $|000\rangle-|111\rangle$, ... [complete to a full orthogonal basis in some appropriate way]?

The paper never even defines this key concept, explicitly enough for a theoretical computer scientist like me to be able to understand it. (It does say that these are measurements that "arise when measuring total angular momentum," which *might* uniquely define what's being talked about to a physicist?) And I didn't feel like tracing back through a chain of earlier papers in search of the answer -- it was THIS paper's job!

(2) The introduction never makes explicit contact with the whole sequence of works on

measurement-based quantum computing (MBQC) that started with Raussendorf and Briegel more than 20 years ago, even though this seems like an obvious point of reference for orienting the reader. In those works, one gets universality for *1-qubit* measurements, never mind the 2-, 3-, and 4-qubit measurements that the rest of the introduction talks about! On the other hand, it only works if the initial state is already something highly entangled, such as a "cluster state." Is the need for an entangled initial state ONLY drawback of the 1-qubit measurement universality theorem, compared to this paper's result? Or is the need for adaptivity (i.e. classical side computation) an additional drawback? Please explain!

(3) The key technical step seems to be to use the singlet/triplet measurements to prepare a close approximation to something called a "Haar-twirled qubit ensemble." After that, one can get universality by simply appealing to a previously-known result. Once again, though, equation (1) did not define the "Haar-twirled qubit ensemble" explicitly enough for me to be able to understand what it is. Is it just the maximally mixed state over the symmetric subspace of three systems of N qubits each? If so, please say that!

(4) I didn't understand AT ALL what it means that these states and measurements are "real-valued in every choice of reference frame," even though that's presented as a central motivation for the results. In the context of a bare collection of qubits, with no spatial let alone spacetime structure imposed on them, what even IS a reference frame? What does it MEAN for these states and measurements to be "real-valued in every choice of reference frame"? How can we see that they in fact are?

If and when the manuscript is revised to explain the four points above, I'd be happy to take another look.

Reviewer #2 (Remarks to the Author):

This paper resolves an open question posed by Hastings and Freedman in 2021. In their 2021 paper, Hastings and Friedman proposed a model of quantum computation in which one only makes singlet-vs-triplet measurements, and the initial state is invariant under global rotations. Such a model of quantum computation has the very interesting property of

being invariant under rotational changes of reference and impervious to error due to global rotations. However, Hastings and Freedman were unable to prove that this model can execute universal quantum computation (i.e. that it implements BQP). In this manuscript the authors have proven this conjecture, and moreover, the proof is concise. I find also that the paper is clearly written and a joy to read. I would recommend that the paper can be published in Nature Communications without further revision. The only suggestion I have for an optional revision is to the title. It seems to me that the title, while formally accurate, emphasizes different aspects of the model than does the main text of the paper.

RESPONSE TO REVIEWER REPORTS

We thank the reviewers for their time and comments. Below the reports are reproduced verbatim (albeit with different formatting). Our responses to each comment are below in red.

Reviewer 1 Comments

The purpose of this paper is to prove that a certain model of measurement-based quantum computation is universal for BQP, thereby resolving an open problem of Freedman, Hastings, and Shokrian-Zini. The model considered is one where the initial state of n qubits is maximally mixed, and where the only allowed operations are measurements that project onto singlet/triplet subspaces. The authors say that this is foundationally interesting, because it shows that universal QC is possible using states and operations that are “always real in every reference frame.”

Is this “just another BQP-universality result” for some particular set of operations, alongside dozens of others that are now known? Or is it indeed of foundational interest and therefore suitable for Nature Communications? Alas, I didn’t feel like I even understood the setup well enough to be able to answer that question with confidence. Here were my main issues:

1. (1) What ARE singlet/triplet measurements? Are they the measurements that project onto

$$|00\rangle + |11\rangle, |00\rangle - |11\rangle, |01\rangle + |10\rangle, |01\rangle - |10\rangle$$

as well as

$$|000\rangle + |111\rangle, |000\rangle - |111\rangle \quad (9)$$

... [complete to a full orthogonal basis in some appropriate way]? The paper never even defines this key concept, explicitly enough for a theoretical computer scientist like me to be able to understand it. (It does say that these are measurements that “arise when measuring total angular momentum,” which *might* uniquely define what’s being talked about to a physicist?) And I didn’t feel like tracing back through a chain of earlier papers in search of the answer – it was THIS paper’s job!

We have addressed this in the new version, explicitly describing the singlet-triplet measurement, which is a particular two outcome projective measurement on two qubits.

2. The introduction never makes explicit contact with the whole sequence of works on measurement-based quantum computing (MBQC) that started with Raussendorf and Briegel more than 20 years ago,

even though this seems like an obvious point of reference for orienting the reader. In those works, one gets universality for *1-qubit* measurements, never mind the 2-, 3-, and 4-qubit measurements that the rest of the introduction talks about! On the other hand, it only works if the initial state is already something highly entangled, such as a “cluster state.” Is the need for an entangled initial state ONLY drawback of the 1-qubit measurement universality theorem, compared to this paper’s result? Or is the need for adaptivity (i.e. classical side computation) an additional drawback? Please explain!

We have put in more discussion on these lines, as we realise that the standard use of ‘measurement based quantum computation’ is in the context of cluster states and related schemes. However, a difference here is that we are not assuming a previously given multi-party entangled quantum state upon which measurements are performed. In the STP=BQP setup the s/t measurement does *everything*, including generating entanglement, dynamics, and extracting information, given (almost any) supply of starting qubits, which may even be largely unknown, and certainly do not need to be entangled.

3. The key technical step seems to be to use the singlet/triplet measurements to prepare a close approximation to something called a “Haar-twirled qubit ensemble.” After that, one can get universality by simply appealing to a previously-known result. Once again, though, equation (1) did not define the “Haar-twirled qubit ensemble” explicitly enough for me to be able to understand what it is. Is it just the maximally mixed state over the symmetric subspace of three systems of N qubits each? If so, please say that!

We have now put in more explanation.

4. I didn’t understand AT ALL what it means that these states and measurements are “real-valued in every choice of reference frame,” even though that’s presented as a central motivation for the results. In the context of a bare collection of qubits, with no spatial let alone spacetime structure imposed on them, what even IS a reference frame? What does it MEAN for these states and measurements to be “real-valued in every choice of reference frame”? How can we see that they in fact are?

We have now put in more explanation of this point. The notion of a reference frame in our context refers to the way in which one picks the x, y, z axes of the Bloch sphere. This might not be the same as choosing spatial axes, depending upon the nature of the

physical qubit. We have elaborated on the connection to reference frames in the discussion section, in a way that will hopefully be accessible to a broader readership.

If and when the manuscript is revised to explain the four points above, I'd be happy to take another look.

Reviewer 2 Comments

This paper resolves an open question posed by Hastings and Freedman in 2021. In their 2021 paper, Hastings and Friedman proposed a model of quantum computation in which one only makes singlet-vs-triplet measurements, and the initial state is invariant under global rotations. Such a model of quantum computation has the very interesting property of being invariant under rotational changes of reference and impervious to error due to global rotations. However, Hastings and Freedman were unable to prove that this model can execute universal quantum computation (i.e. that it implements BQP). In this manuscript the authors have proven this conjecture, and moreover, the proof is concise. I find also that the paper is clearly written and a joy to read. I would recommend that the paper can be published in Nature Communications without further revision. The only suggestion I have for an optional revision is to the title. It seems to me that the title, while formally accurate, emphasizes different aspects of the model than does the main text of the paper.

We have now changed the title to *The two-qubit singlet/triplet measurement is universal for quantum computing given only maximally-mixed initial states* to more closely reflect the focus of the main text.

REVIEWERS' COMMENTS

Reviewer #1 (Remarks to the Author):

This new manuscript is a VAST improvement in clarity over the previous one! I thank the authors for addressing all of my questions, and I now strongly support this excellent paper for publication in Nature Communications.

I only have one tiny comment: in the introduction, the authors should EXPLICITLY say that, by the "s/t measurement," they mean the measurement where one outcome is $(|01\rangle - |10\rangle)/\sqrt{2}$ and the other outcome is its orthogonal complement. This is the central concept of the whole paper, yet the paper still starts talking about "s/t measurements" while leaving it unstated that these are the measurements discussed in a previous paragraph. (If you must know, I was thrown off because I firmly expected a "triplet measurement" to involve 3 qubits.)